# MultiGAS Detection from Airborne Platforms on Italian Volcanic and Geothermal Areas

Malvina Silvestri [1,*](ORCID), Jorge Andres Diaz [2,3], Federico Rabuffi [1,4](ORCID), Vito Romaniello [1], Massimo Musacchio [1](ORCID), Ernesto Corrales [2](ORCID), James Fox [3], Enrica Marotta [1](ORCID), Pasquale Belviso [1], Rosario Avino [1], Gala Avvisati [1] and Maria Fabrizia Buongiorno [1](ORCID)

1 Istituto Nazionale di Geofisica e Vulcanologia, 00143 Rome, Italy; federico.rabuffi@ingv.it (F.R.); vito.romaniello@ingv.it (V.R.); massimo.musacchio@ingv.it (M.M.); enrica.marotta@ingv.it (E.M.); pasquale.belviso@ingv.it (P.B.); rosario.avino@ingv.it (R.A.); gala.avvisati@ingv.it (G.A.); fabrizia.buongiorno@ingv.it (M.F.B.)
2 GasLab, CICANUM, Universidad de Costa Rica (UCR), San José 11501, Costa Rica; andres.diaz@inficon.com (J.A.D.); ernesto.corrales@ucr.ac.cr (E.C.)
3 INFICON Inc., East Syracuse, NY 13057, USA; james.fox@inficon.com
4 Dipartimento di Scienze, Università Degli Studi Roma Tre, 00146 Rome, Italy
* Correspondence: malvina.silvestri@ingv.it

**Abstract:** The measurement of volcanic gases, such as $CO_2$ and $SO_2$, emitted from summit craters and fumaroles is crucial to monitor volcanic activity, providing estimations of gases fluxes, and geochemical information that helps to assess the status and the risk level of an active volcano. During high degassing events, the measurement of volcanic emissions is a dangerous task that cannot be performed using hand portable or backpack carried gas analysis systems. Measurements of gases plumes could be safety achieved by using instruments mounted on UAS (Unmanned Aerial System). In this work, we present the measurements of $CO_2$, $SO_2$, and $H_2S$ gases collected with a miniaturized MultiGAS instrument during 2021 and 2022 field campaigns. They took place at several thermally active areas in Italy: *Pisciarelli* (Naples, Italy), *Stromboli* volcano (*Messina*, Italy), and *Parco Naturalistico delle Biancane* (*Grosseto*, Italy).

**Keywords:** volcanic gases; miniaturized instruments; geothermal areas

## 1. Introduction

Characterization of gas emissions at dangerous or difficult to reach areas (i.e., geothermal areas or active volcanoes) without risking injuries or even the lives of scientists and personnel performing the analysis during field campaigns can be enhanced with the use of miniaturized instruments mounted onboard Unmanned Aerial System (UAS) [1].

In this study, we report results of plume investigations at Italian sites, using a small MultiGAS sensing payload (*miniGAS*) designed for either hand-portable, ground and airborne platform operations. The measurements of $CO_2$, $H_2S$, and $SO_2$ were collected both hand carrying the instrument and mounting it on the UAS. The selected areas were Italian thermally active sites (Figure 1) that were well studied in terms of gases emission: *Pisciarelli* area (at NE of the *Solfatara* volcano, Napoli) and *Stromboli* Island are characterized by volcanic activity, while *Parco Naturalistico delle Biancane* (*Grosseto*) presents thermal manifestation with fumaroles in a geothermal context. While particular attention is given to the two active volcanic areas (*Pisciarelli* and *Stromboli*) with a constant monitoring of gases measured by ground stations and dedicated field campaigns [2–4], *Parco Naturalistico delle Biancane* is characterized by a geothermal field with hot and pressurized fluids used for the production of electricity [5,6] and requires gases measurements for geothermal system characterization. For these reasons, the measurement of the gases needs more attention for both volcanic surveillance and geothermal prospection (Figure 2). Specifically, close

to *Pisciarelli,* there are numerous inhabited centers which could be interested by gases exhalations; volcanic activity of *Stromboli* is usually characterized by violent eruptions that can be dangerous for citizens and tourists; geothermal area of *Parco delle Biancane* is continuous interested by underground phenomena manifested with gases emissions and heating of the surface.

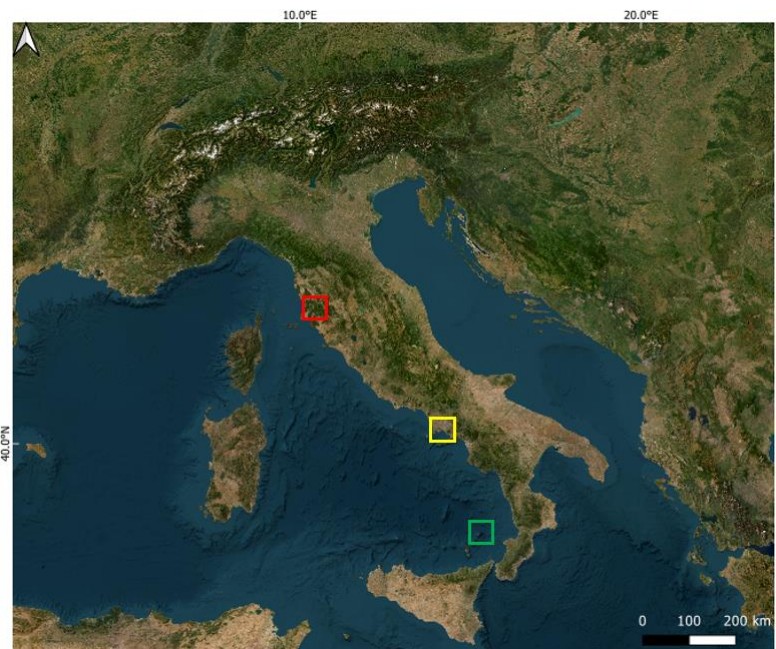

**Figure 1.** Test sites location: *Pisciarelli* area (yellow box), *Stromboli* Island (green box) and *Parco Naturalistico delle Biancane* (red box). Basemap from ESRI satellite (World_Imagery (MapServer)—ArcGIS).

The gas content information obtained from in situ measurements or from UAS systems is also useful to assess and validate satellite retrievals [7,8]. The integration of satellite and in situ collected data can be used, for example, to improve retrieval of SO$_2$ emissions from craters.

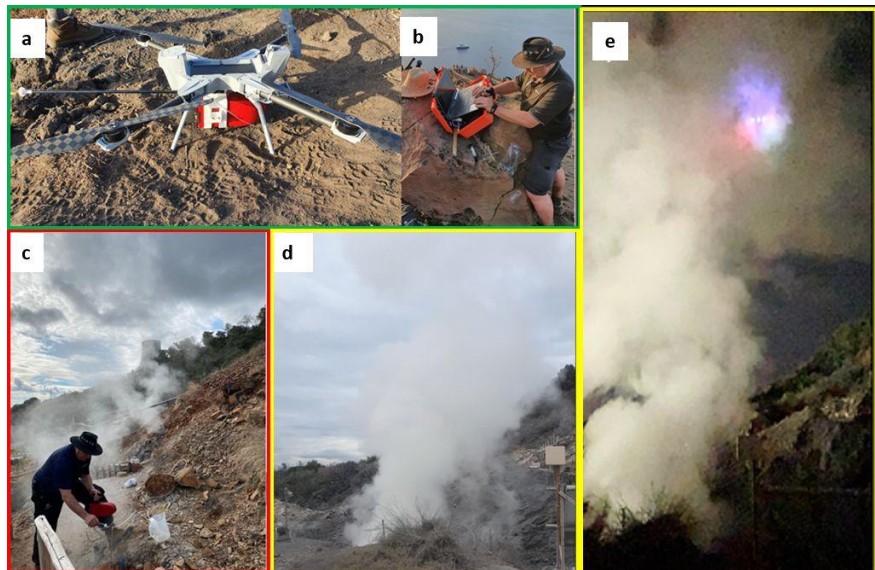

**Figure 2.** Field campaigns: (**a**) *miniGAS* mounted on SR-X4 [9] UAS during the *Stromboli* field campaign; (**b**) real-time remote data collection (1 km from the summit); (**c**) *miniGAS* at the *Parco Naturalistico delle Biancane* site hand carried during ground survey; (**d**) *Pisciarelli* biggest fumarole; (**e**) *miniGAS* mounted on SR-X4 UAS collecting data of gas concentration during the night (*Pisciarelli*).

## 2. Materials, Methods, and Sites Description

### 2.1. Miniature MultiGAS Sensing Payload

The *miniGAS* instrument (Figure 3) was developed by Dr. Jorge Andres Diaz and includes temperature, pressure, relative humidity, $SO_2$, and $H_2S$ electrochemical sensors and a non-dispersing near infrared $CO_2$ sensor. Moreover, it also includes a GPS sensor, onboard data storage, and telemetry via mid-range RF antennas with up to 1.5 km of connectivity range for real-time and remote in situ gas data collections. The new version (*miniGAS PRO*) deployed during the 2022 Italian campaigns consists of a Campbell CR310 data logger recording signals from a PP System $CO_2$ infrared spectrometer (SBA-5 OEM; 0–2000 ppm concentration range), which also includes a solid-state $H_2O$ partial pressure sensor (0–40 mbar range). The data logger also records signals from up to four City Technology (UK) electrochemical sensors: two for $SO_2$ (EZT3ST/F; 0–200 ppm; 2TD2G-1A 0–10 ppm concentration range), one for $H_2S$ (0–50 ppm; 2TC4E-1AEZT3H; 0–50 ppm; concentration range), and one optional $H_2$ sensor T3HYT; 0–100 ppm concentration range). Gas enters the inlet located 1.2 m away from the drone core. The distance was selected to reduce the effects of the rotor draft flow over the plume to sample "undisturbed plume gas" as much as possible. The gas sample is circulated through the system through ¼ PTFE tubing by a small diaphragm pump at a flowrate of ~1.2 lpm, which connects the inlet to the sensor array. A 1.2 um PTFE Teflon filter is attached to the tubing as an inlet to avoid dust, particles, and small droplets into the *miniGAS*. Moreover, information on time, temperature, pressure, relative humidity, and position are also recorded with the gas concentration data and prepared to be send by telemetry by the data logger. Data is both recorded and transmitted at 1 Hz from the UAS to a laptop computer via radio transmitter (RF-407 RPSMA radio transmitter from Campbell Scientific at 915–928 Mz.). The *miniGAS* has a universal adapter for easy and quick integration to different UAS platforms.

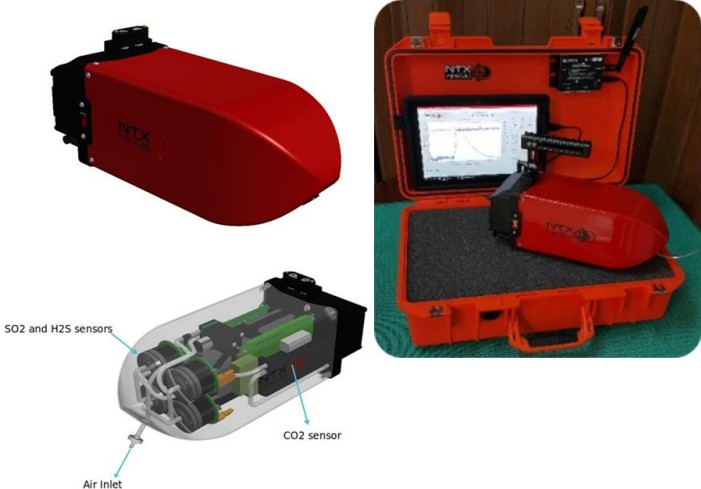

**Figure 3.** *miniGAS PRO* System.

The weight of the *miniGAS PRO* version is 1.5 kg, this includes the rechargeable LiPo battery (which allow up to 2 h of operation) and the weatherproof glass/carbo fiber aerodynamic case. The size is 28 cm L × 12 cm H × 12 cm W), which makes it very compact for multirotor or fix wing drone operation. Earlier versions of the *miniGAS* (Alpha, Beta) have been flight tested and deployed in Costa Rica within the Turrialba volcano plume [1,10–13] using different drones and tethered balloons platforms, generating real time 3D gas concentration plots of active volcanic plumes. The *miniGAS PRO* is a newer version from the earlier *miniGAS* NTX deployed during the *Solfatara-Vulcano* 2015 campaign to characterize fumarolic sites [14]. Improvements include better electrochemical and infrared sensors, more robust data logger, lighter frame, ruggedized fiber glass/carbon case for harsh conditions, new GPS, new PCBs, and better software-hardware interfaces. This newer version

has been successfully deployed within active volcanic plumes at Turrialba and Masaya volcanoes [15–17], Mt. Etna in 2018, Poas volcano [18], Kilauea volcano [19], Manam and Tavurvur volcanoes in 2019 [20], and at an industrial field gas sensing application in Scotland and Kazakhstan in 2019.

The measurements were conducted either by hand carrying the instrument into the fumaroles ("sampling walk") or by flying different UAS over the fumarolic sites. Each gas sensor is calibrated prior to each deployment using calibration gases (Zero and Span gas concentration values for each sensor). Following this approach, the system is exposed to the calibration gases via sample bag connected to the inlet and once stable, the analog signals are adjusted to the know concentration to provide in real time calibrated concentration data. Time response (system response time and sensor response time) is incorporated as part of the calibration process of the miniGAS, so concentration data reflect exact GPS position and exact concentration values when sampling for a short period of time. Raw data (Digital, voltage and mA signals) are also recorded and transmitted during the field measurements for post field deployment calibration if necessary.

### 2.2. Unmanned Aerial System (UAS)

The SR-X4 system is a multi-rotor drone designed to optimize performance by reducing weight and dimensions, in order to provide a multi-role and multipurpose instrument with high performance in terms of operating range and autonomy [9].

SR-X4 quadcopter UAS has an extended flight endurance and range, allowing to fly up to 40 min, 15 km away, and carrying a payload of up to 2 kg. Table 1 reports the main features. The SR-X4 System is a multi-role system with great versatility, able to maximize performance in flight and range. It is extremely accurate and quiet, with fast actuation times and it features a high load capacity. This UAS is the ideal platform to integrate specialized and multi-sensor payloads, suitable for missions that require a large operational range and high persistence on the area of interest. The entire system can be easily transported in a practical fly case or even in a tactical backpack compliant with the "Iata" standard. It also has the possibility to integrate a thermal payload (Figure 4).

**Table 1.** UAS SR-X4 main technical characteristics.

| Description | Value |
| --- | --- |
| Endurance | Up to 40 min |
| Range | 15 km |
| Payload capacity | up to 2 kg |
| Wind tolerance | Up to 28 knots |
| Climb rate | Up to 5 m/s |
| Speed | 20 m/s (manual); 12 m/s (GPS) |
| Case | Waterproof IPX5 |
| Temperature | −20 °C to 45 °C |
| Materials | Single fiver carbon frame. Fiver carbon blades and rods. |

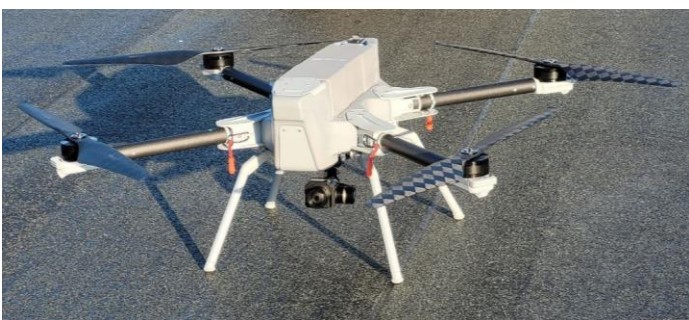

**Figure 4.** SR-X4 quadcopter UAS.

## 2.3. Field Campaigns

Three field campaigns were conducted in different sites (Table 2).

**Table 2.** Sites where *PRO* sensor was used. The third column report the modality of data acquisition.

| Site | Date | UAS/Ground |
|---|---|---|
| *Pisciarelli* | November 2021, July 2022 | UAS |
| *Parco Naturalistico delle Biancane* | May 2022 | Ground (hand) |
| *Stromboli* | September 2022 | UAS |

*Pisciarelli* fumarole is one of the main fumaroles located at the NE part of the *Solfatara* area (Naples) [21–24]. This fumarolic field is about 0.03 km$^2$ fault-related hydrothermal area and has shown clear signs of increasing hydrothermal activity during the ongoing phase of unrest. *Pisciarelli* is characterized by the presence of large mud pools with intense bubbling, and by high flow rate fumaroles emitting a $H_2O$-$CO_2$ rich gas mixture with minor $H_2S$ [23]. In recent years, the fumarolic field of Pisciarelli has experienced an evident increase in activity, which has been marked by temperature increases, by the opening of new vigorous vents and boiling pools, and by the occurrence of seismic activity localized in the area [21]. Fumaroles emit vapors characterized by the absence of acidic gas species (i.e., $SO_2$, HCl, and HF) that are typical of high-temperature magmatic gas emissions. They also exhibit detectable amounts of species formed in the hydrothermal environment (i.e., $CH_4$). These compositions suggest fumarolic steam originates from a hydrothermal system [22].

The *Parco Naturalistico delle Biancane*, located in Southern Tuscany, belongs to the *Larderello* geothermal field [25]. It is characterized by the presence of geothermal surface manifestation (i.e., hot spring, mud pool, fumarole, and soil alteration). The *Larderello-Travale* geothermal system is exploited for energy production with about 30% of the Tuscany energy demand [5,6]. The *Larderello* geothermal area is a magmatic-driven geothermal system located in the inner Northern Apennines [25]. The *Parco Naturalistico delle Biancane* is located in the southern sector of the *Larderello-Travale* geothermal field. It extends over an area of approximately 100,000 m$^2$. Previous studies on *Parco Naturalistico delle Biancane* test site demonstrated that the fluids released consist mostly of water at its boiling point and are primarily composed of $H_2O$. Other components include $CO_2$, $N_2$, $H_2S$, $H_2$, and $CH_4$, with trace amounts of $O_2$ and noble gases [26].

Continuous degassing and moderate explosive eruptions characterize the ordinary (or background) activity of the *Stromboli* Volcano [27–30]. For the *Stromboli* volcano, the gases observations are collected from safe locations, hundreds of meters from the erupting vents. The summit of *Stromboli* volcano, the northernmost island of the Aeolian volcanic arc (Italy), reaches 924 m in altitude. The island represents the top part of a large 2500 m high stratovolcano emerging from the Tyrrhenian Sea floor. In this context, gaseous volcanic emissions consist of a variety of different compounds and are dominated by water vapor ($H_2O$), carbon dioxide ($CO_2$), sulfur dioxide ($SO_2$), and hydrogen sulfide ($H_2S$). The ratio of these gases varies depending on the type of eruptive activity [31].

The *Istituto Nazionale di Geofisica e Vulcanologia* (INGV) is responsible for the seismic and volcanic surveillance through the installation and management of monitoring networks, technologically advanced, in order to monitor the entire national territory and the Mediterranean basin. In recent decades, the use of drones equipped with sensors to measure the gases emitted by active volcanoes demonstrated that even in inaccessible, active and dangerous volcanoes, is the only way to implement important measures to characterize the state of activity in safe conditions. INGV fleet counts about 40 aircrafts with visible and thermal cameras and the *miniGAS* payload to measure volcanic gases [32].

## 3. Results

### 3.1. Pisciarelli

During the field campaign held on November 2021 and July 2022, at *Pisciarelli* area, gases measurements were collected by mounting the *miniGAS* on SR-X4 drone. The aim of November 2021 campaign was to test the *miniGAS* measuring the volcanic gases present in the main fumarolic plume to better understand its geochemistry. Figure 5 shows the flight path of the UAS for both missions. The colors vary according to the $CO_2$ concentration, from green (background level values) to red (high concentration values). The evident saturation of $CO_2$ (Figure 5) is because the measurements were collected when the drone went into the volcanic plume.

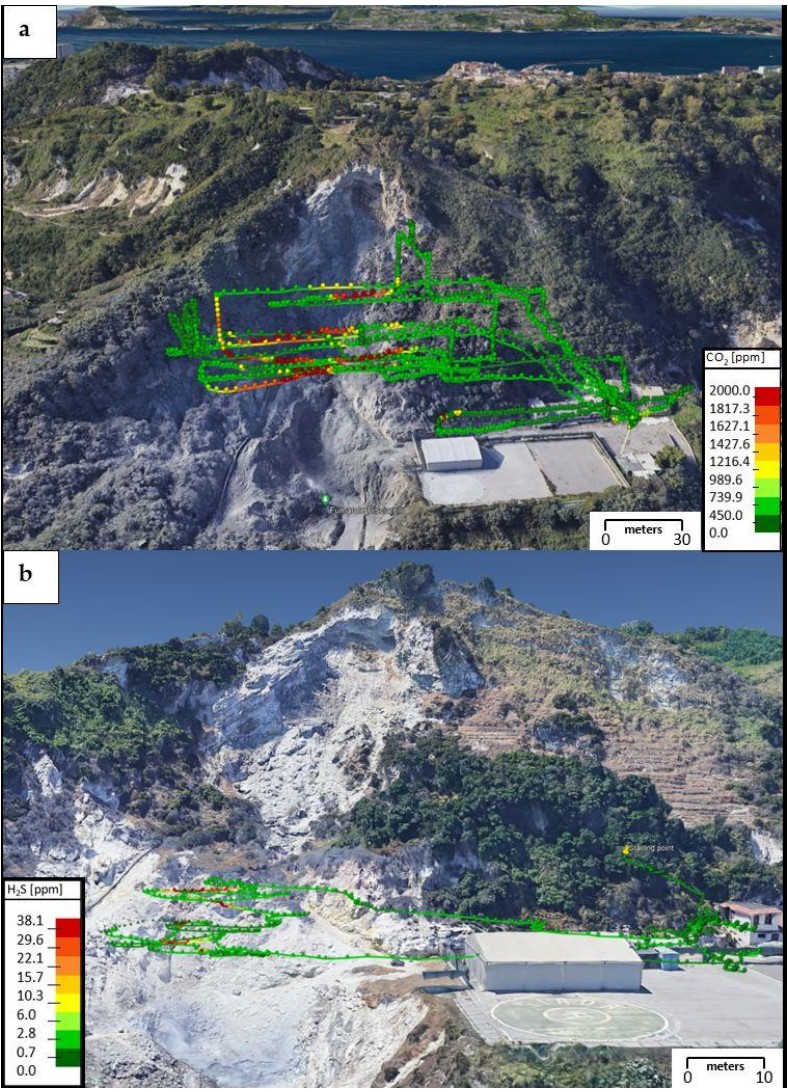

**Figure 5.** *Pisciarelli*: (**a**) November 2021 path of the drone and $CO_2$ concentration. Green to red colors indicate increasing values of the gas concentration. (**b**) July 2022 $H_2S$ concentration taken by UAS. Overlaying maps from Google Earth.

By analyzing the data, no $SO_2$ concentration in accordance with [23]; however, significant concentrations of $CO_2$ and $H_2S$ are present. The major fumarolic fluid gases in a hydrothermal system, excluding $H_2O$, are $CO_2$ and $H_2S$.

Moreover, the $CO_2/H_2S$ ratio calculation helps to establish a common reference point and to maintain coherence in measurement (Figure 6).

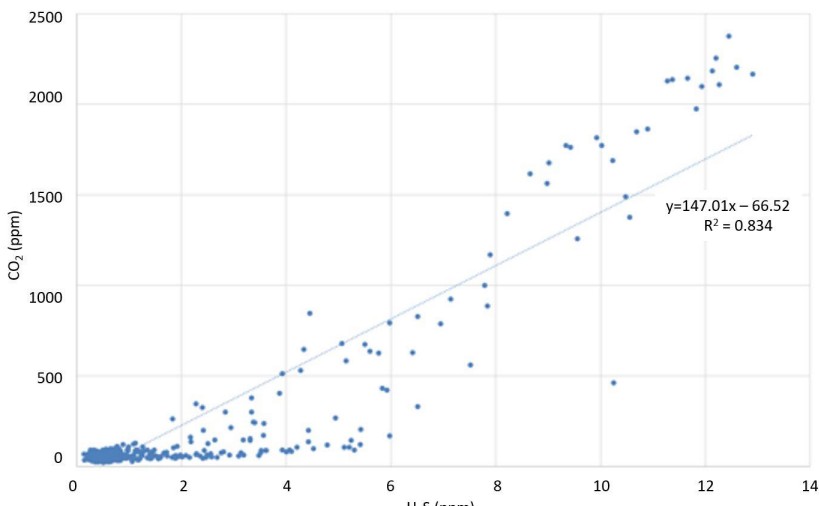

**Figure 6.** $CO_2$ vs. $H_2S$ concentration (ppm) scatterplot. The $CO_2/H_2S$ ratio is represented in blue circles. Line represents the best-fit regression.

All $CO_2$ concentrations are corrected for the atmospheric background (about 410 ppm, which is consistent with the European Centre for Medium-Range Weather Forecasts, ECMWF model estimations [33]). The $CO_2/H_2S$ ratio value of about 147 in *Pisciarelli* indicates that the fumarole is dominated by carbon dioxide and has very low levels of hydrogen sulfide. The ratios are independent of sampling altitude and distance to the fumarole.

This measured value of the $CO_2/H_2S$ ratio is in line with the same ratio measured at the fumaroles of the *Solfatara* in recent years [24].

### 3.2. Parco Naturalistico Delle Biancane

The gas data measurements at *Parco Naturalistico delle Biancane*, (May 2022) were collected by hand-carrying the *miniGAS*. The distance between the gas systems and the ground was approximatively of 20 cm and this permitted a very detailed analysis of the fumaroles present on site (Figure 7).

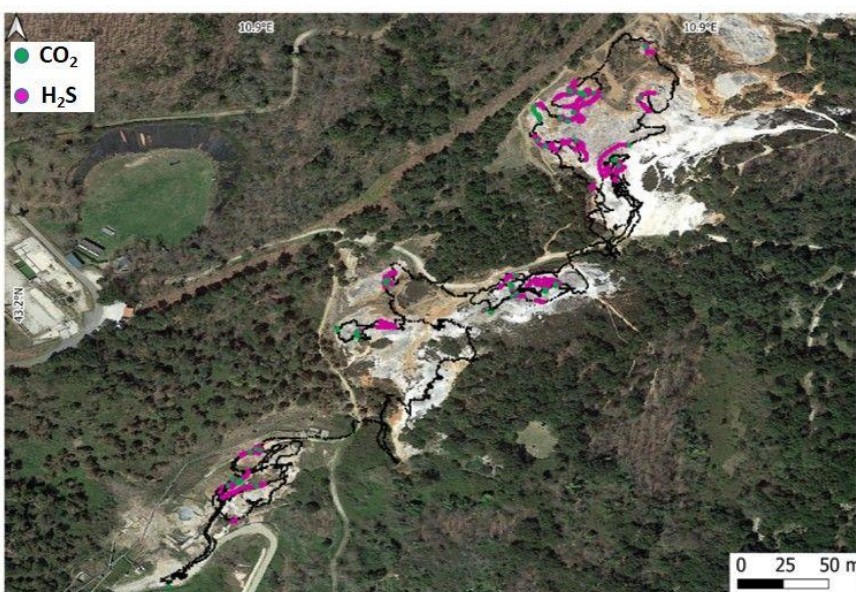

**Figure 7.** *Parco Naturalistico delle Biancane*: $CO_2$, $H_2S$ and path of the *miniGAS* ground measurements; location of measurements (black), peaks of $H_2S$ (green), and $CO_2$ concentration peaks (>600 ppm, purple). Basemap from Google Satellite (World_Imagery (MapServer)—ArcGIS).

By comparing the measurements collected during the May 2022 field campaign and the ones collected by [34], we can consider that:

(1) Fumaroles are located close to the ones detected by *miniGAS* (Figures 7 and 8);
(2) Despite the different period, instrumentation and methodology used by the authors to measure $CO_2$ and $H_2S$, in respect to ours, concentration values are similar and comparable (Table 3). Specifically, Table 3 reports the $CO_2/H_2S$ ratio for two representatives fumaroles (B4 and B6) measured by [34] and *miniGAS*, respectively.

**Table 3.** Comparison of $CO_2/H_2S$ ratio between [34] and *miniGAS* measurements.

| Methodology | B4 | B6 |
|---|---|---|
| Granieri et al., 2023 [34] | 40.6 | 34.1 |
| *miniGAS PRO* | 29.9 | 33.3 |

The results obtained during the May 2022 field campaign seems to be consistent with measurements achieved by [34].

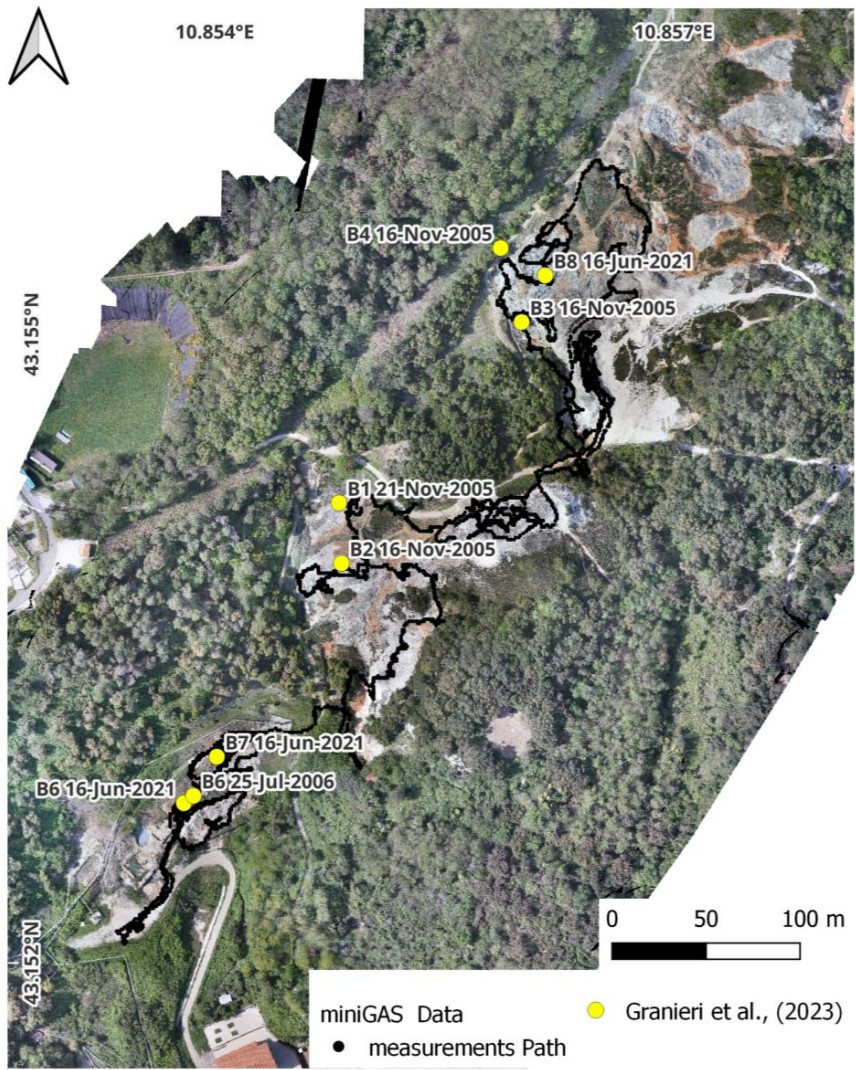

**Figure 8.** Main fumaroles measured by Granieri et al. [34] (circle in yellow with name and date of measurement) over *miniGAS* path measurements. Visible image collected by camera mounted on during the May field campaign.

### 3.3. Stromboli

In the frame of FIRST (ForecastIng eRuptive activity at Stromboli volcano: timing, eruptive style, size, intensity, and duration) project, a field campaign on *Stromboli* volcano was organized with the aim to collect information on gases concentration emitted by the volcano. The project, in fact, is aimed at improving the ability of the scientific community to forecast the Stromboli behavior in terms of changes in the eruptive style and scale of the future expected eruption, basing on the monitoring data and on geophysical and geochemical observables. In particular, it is aimed at defining the state of the volcano, distinguishing between background activity and exceptional eruptions (such as violent explosions, paroxysms, or flank eruptions), combining together the monitoring data from the INGV networks, satellite data, and the numerical modelling of gas and magma propagation [35].

During the field campaign held on September 2022, the first in situ measurements of a volcanic plume from *Stromboli* summit craters, using the *miniGAS* chemical instrument integrated into the SR-X4 UAS, were collected. The UAS was piloted by hand inside the volcanic plume at different heights. The instrument on-board measured $CO_2$, $SO_2$, $H_2S$, and $H_2O$ gases together with GPS position.

Figure 9 shows the in situ measurements of $CO_2$ and $SO_2$ gas concentration taken at the active degassing plume of *Stromboli* volcano.

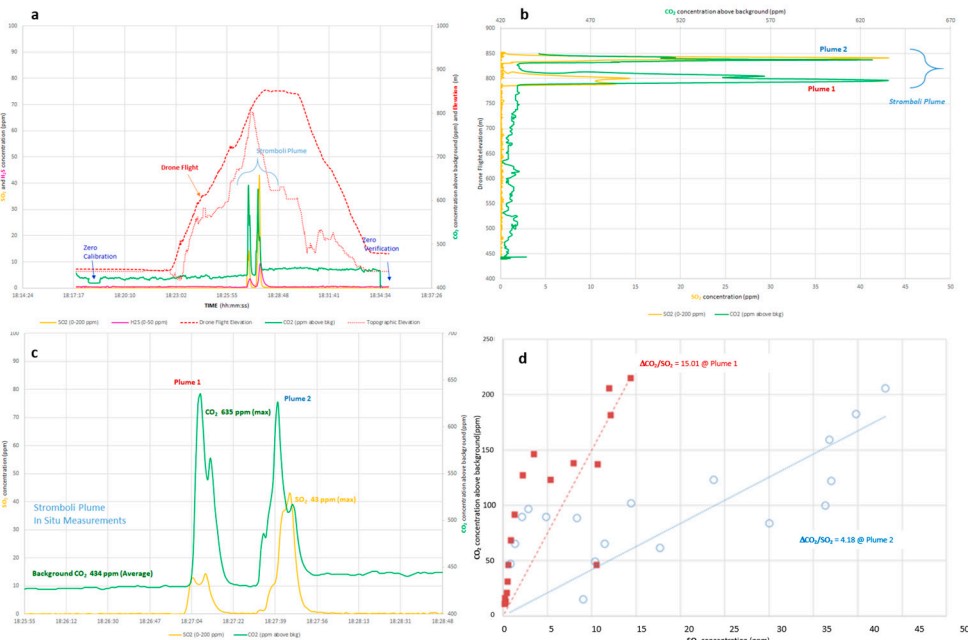

**Figure 9.** *Stromboli* volcano field campaign results using SR-X4 + *miniGAS PRO*. (**a**) Measured gas concentrations of $CO_2$, $H_2S$, $SO_2$ and drone altitude (in meters) during drone flight; (**b**) Altitude vs. $CO_2$ and $SO_2$ to provide plume height and thickness; (**c**) Detailed $CO_2$, $SO_2$ concentration vs. time; (**d**) $CO_2/SO_2$ ratios for Stromboli measured plumes. The red squares represent the $CO_2/SO_2$ ratio for the Plume 1, blue circles the $CO_2/SO_2$ ratio for the Plume 2 and the lines the best-fit regression for Plume 1 and Plume 2 (red and blue respectively).

The *miniGAS* mounted on UAS flew through the active plume two times, which were located between 800 and 850 amsl. The volcano presented small eruptions during the gas data collection. Two distinct plumes are observed, the first one (denoted Plume 1 in Figure 9b,c), with a high content of $CO_2$ concentration above background levels is 15 times higher than the $SO_2$ concentration. Plume 2 (Figure 9b,c), located 50 m higher than Plume 1, has a smaller $CO_2/SO_2$ ratio, $CO_2$ above background concentration is just 4.2 times the $SO_2$ content, which might be indicative of the variability and source and location of each gas emitted plume.

Unfortunately, because of a wildfire occurred in May 2022 (which destroyed the vegetation) and mudslides three months later, the accessibility to the summit area was very dangerous; for this reason, the "*home*" point for the UAS was chosen away from the crater, which is a safer location during Strombolian activity (Figure 10).

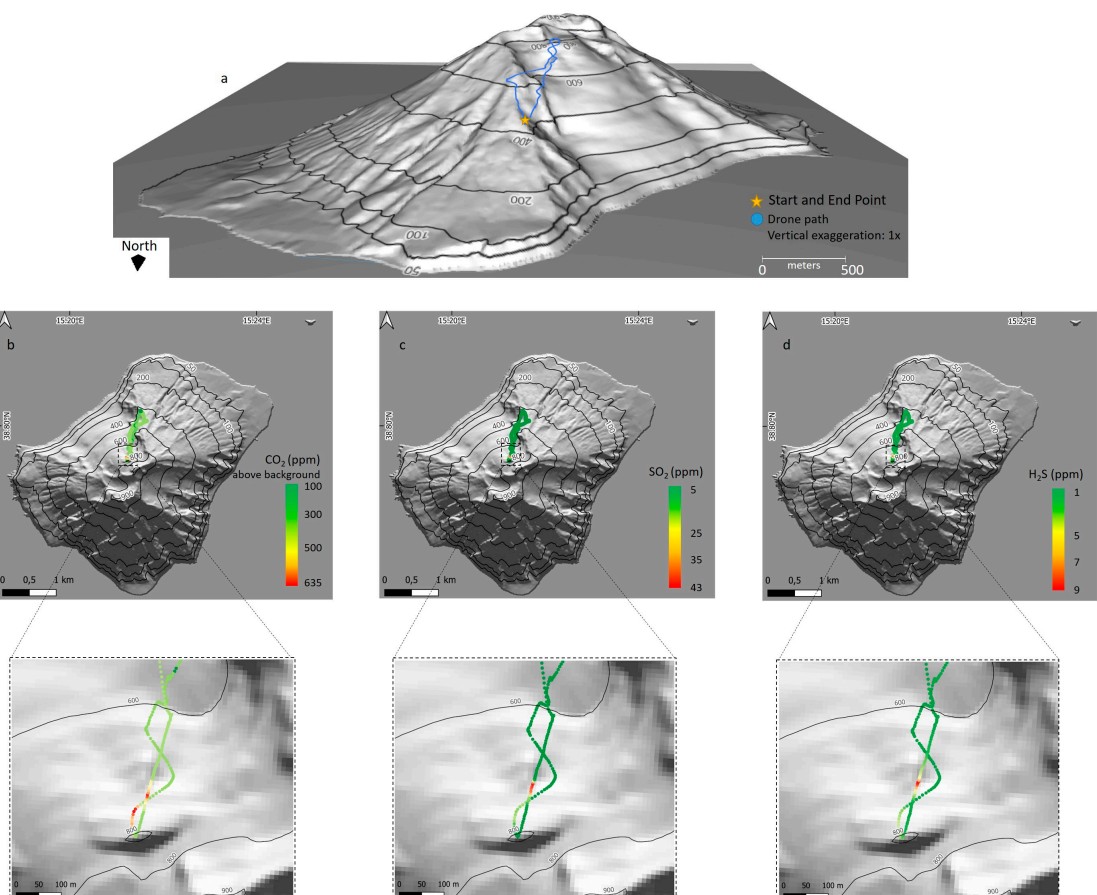

**Figure 10.** *Stromboli* volcano: (**a**) path of the drone overplot on DEM at 10 m of spatial resolution [36]; gases concentration with their zoom: (**b**) $CO_2$ gas concentration visualization (green to red colors indicate increasing values of the concentration), (**c**) $SO_2$ gas concentration map, (**d**) $H_2S$ gas concentration map.

## 4. Discussion and Conclusions

In this paper, we have demonstrated that MultiGAS systems, such as the *miniGAS PRO* version, carried by drones is a viable option for systematic gas measurements of volcanic emissions. It offers a robust system for the monitoring of volcanic activity using UASs but can also be hand carried for a more detailed monitoring when conditions are favorable to visit the fumarolic site.

The gas instrumentation, camera manufactures, and drone industry have evolved to provide small, simple, and easy to deploy volcanic UAS for routine and rapid respond field campaigns. INGV has implemented a capability to routinely monitoring the Italian volcanoes with the UAS to measure gases and map active areas to correlate with ground and satellite base measurements.

Active sites, such as *Pisciarelli*, *Parco Naturalistico delle Biancane*, and *Stromboli,* but also other Italian volcanoes have been monitored using UAS by INGV. These UAS measurements provide crucial parameters in terms of geochemical data and mapping information to assess the status and the risk level of an active volcano and monitor the development of volcanic activity without risking INGV personnel. Same or similar instrumentation can be implemented at volcanic observatories around the world to monitor volcanic activity at a



distance to provide a quick and safe risk assessment. More advances in the area of payload and UAS developments are underway to further impact the field of volcanic research and monitor.

Future work will concentrate in the improvements of the *miniGAS* payload including more sensors to detect and retrieve other emitted gases characteristics (such as $CH_4$). Moreover, the capability to mount the thermal cameras and the gas sensors at the same time onboard the UAS will be analyzed. This upgrade will be tested on these three test sites and other volcanic and thermal areas in Italy (i.e., *Vulcano* and Mt. Etna).

Another area of work is the estimation of volcanic $CO_2$ from remote sensing and in situ validation. In situ measurements of carbon dioxide emissions could be useful for the validation of $CO_2$ enhancements ($XCO_2$) from satellite. In fact, the availability of hyperspectral images from satellite is very challenging for gases analysis. Specifically, the PRISMA (PRecursore IperSpettrale della Missione Applicativa) Italian mission provides hyperspectral images in the range 400–2500 nm with a spatial resolution of 30 m [37]. The spectral range includes the $CO_2$ absorption bands around 2000 nm in the SWIR (Short Wave Infra-Red) region. In this framework, the development of algorithms and employment hyperspectral data to estimate $CO_2$ concentrations inside volcanic plumes is constantly evolving [38]. Although satellite data have a huge potential to cover large areas and for safety of the researchers, gases emissions are not so easy to detect and estimate. Our objective is the comparison between satellite and *miniGAS* measurements in order to be confidence on the considered retrieval methodologies. While satellites can readily detect and measure $SO_2$ emissions [39], estimations of volcanic $CO_2$ emissions are more challenging because the high background concentration in the atmosphere (about 400 ppm). The only way to get accurate readings is to take samples near active vents. For this reason, next field campaigns will be planned in order to collect data in synchronous with satellite passages. Considering that sensors aboard satellites measure gases in the total atmospheric column (this means the entire vertical column of air between the satellite and ground), it is fundamental to design the flight path of the drone to obtain gases fluxes estimations from the vents with the aim to make local measurements comparable with satellite estimations.

**Author Contributions:** Conceptualization, M.S. and J.A.D.; methodology and software, J.A.D.; validation, M.S., R.A., V.R. and M.M.; drones, E.M., P.B. and G.A.; data curation, F.R. and E.C. Sensor Calibration, J.F.; writing—original draft preparation, M.S. and J.A.D.; funding acquisition, M.F.B. All authors have read and agreed to the published version of the manuscript.

**Funding:** Part of this research was funded by the Project FIRST-ForecastIng eRuptive activity at Stromboli volcano: timing, eruptive style, size, intensity, and duration, INGV-Progetto Strategico Dipartimento Vulcani 2019 (Delibera n. 144/2020). UCR funded GasLab-CICANUM activities under Research Project #915-C0-116 (VAMOS-UAS) through the Vicerrectoria de Investigacion. INFICON contributed with field travel funding for J.A.D. and instrument calibration at ISS facilities.

**Data Availability Statement:** Data collected by *miniGAS PRO* are available by contacting the correspondence author.

**Acknowledgments:** Special thanks to Giuseppe De Rosa, volcanological guide for its help in the Stromboli field campaign. The authors wish to thank the reviewers for their constructive comments.

**Conflicts of Interest:** The authors declare no conflict of interest.

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
