# Peer review of "MultiGAS Detection from Airborne Platforms on Italian Volcanic and Geothermal Areas"

_remotesensing, doi:10.3390/rs15092390_

Round 1
Reviewer 1 Report
Dear Authors,
This paper is a well-written technical report on the detection of multigas systems carried by drones, which are shown to be a viable option for systematic gas measurements of volcanic emissions in dangerous areas. You took three different case studies in Italy showing the application of this tool.
I have very little to add to evaluate the scientific resonance of the paper although I think you would agree that this paper needs to reinforce the comparison with hand-made measurements on the same areas, by taking into consideration a wide sample statistic, that, if possible, embraces a large span of time and more authors (Cf. Table 3). I would contextually reinforce the geological aspect to show that the application of this new tools in the last years has allowed to definitely improve the capacity of predicting geological events associated with the change of values measured on volatile emissions.
From a geological perspective, I think the geographic location figures (es. Fig. 7 and 10) have little to add to the story and may be restructured accordingly to eventual changes to the paper. Also Fig. 5 and 12 were possibly downsampled in the transmission of the review pdf file, please check it out.
In the attached file, I add hereby some very minor comments to the paper.

Author Response
Thanks for your comments and suggestions. In attached file our replies to your revisions.

Reviewer 2 Report
General comments:
The technical note reports the results of plume investigations at three Italian sites, carried out using a new version (miniGAS PRO) of a small multigas sensing payload (miniGAS) designed for either hand-portable, ground and air-borne platform (UAS systems) operations. The authors also propose an attempt to use the gas content information obtained from UAS systems in order to assess and validate satellite retrievals.
The authors express clearly enough the purposes of the note and its applications as well as its novelty. The topic has been addressed in an appropriate manner and in the context of previous literature, describing the hypotheses and results in terms of the current state of the field.
In my opinion a major flaw is not having provided information on the response times of the sensors. It is known that any differences between these times introduce uncertainties in the ratios between the species, which are amplified for the instruments installed on UAS.
The Authors describe the results of an acquisition of hyperspectral satellite images on Stromboli volcano within the Italian PRISMA mission and report the testing of retrieval algorithms based on CIBR techniques. They also highlight the limits of estimating CO2 emissions using this methodology and conclude by enunciating some good practices for making drone measurements comparable with satellite ones. Although this approach is appreciable, however the lack - in this paper - of a comparison between the two methods, makes this paragraph the mere enunciation of a future purpose.
I found figures and tables very powerful and illuminating. I see no other major flaws, except some minor highlighted below. Please, always specify the source of aerial images, when they are not taken from the camera of the drone (e.g. Google maps). In fig. 10 the shape of the island of Stromboli seems deformed: please check.
In some cases, both in the text and in the references, chemical compounds are indicated without subscripts (e.g. in line 177 or 351). Please check and correct.
The English is good enough for the most of the parts; however, although I am not an english-native speaker, feel that many sentences need to be improved. Thus, I would strongly recommend a thorough revision by a professional proofreading. I took the liberty of suggesting some modifications and minor corrections to the text in order to make it more effective, in my view. Feel free to accept them as they are or take a cue from them.
Discussion and conclusions seems to be written somewhat hastily, under great time pressure. It is a real pity that the conclusions of the work do not give enough value to the results obtained, which to me seem very good. Similarly, it would be appropriate to better highlight the critical issues encountered regarding the measurements made and the possible solutions.
My final recommendation is that the paper can be accepted for publication in the journal after minor revision of the manuscript that will make it more suitable for the publication. If necessary, I will be very glad to review the manuscript after revision.
Detailed comments
Abstract
Lines 24-25: Please rephrase according to the suggestions given for chapter 4.
Introduction
Lines 45-46: It is a rather generic statement: please rephrase it by contextualizing it in the topic of the paper.
2. Materials, Methods and sites description
2.1 Miniature Multigas Sensing Payload
Lines 72-74: Rotor-driven atmospheric mixing results in spatially averaged sampling of poorly constrained plume volumes with implications for the discrimination of closely spaced emission sources. Maybe 1.2 m distance of the gas inlet from the drone is just a few to say "to avoid air disturbance due to the rotor". Unless you are absolutely certain, perhaps it would be better: "in order to reduce air disturbance". In any case it would be useful to explain why a distance of 1.2 m was chosen and to cite appropriate references.
2.3 Field campaigns
Line 131: Perhaps “Pisciarelli is one of the main fumaroles in the Solfatara area” would be better than: “The Pisciarelli fumarole is the strongest in the Solfatara area”
Line 142: It would be more correct to write that "… characterizes the ordinary (or background) activity of the Stromboli Volcano."
Line 145: Perhaps "is responsible for" would be better than: "is in charge of organizing"
3. Results
Each of the paragraphs into which the results chapter is divided contains, in addition to the acquired data, a description of the three sites in which the measurements were carried out. In my opinion it would be more appropriate to move the description of the sites to paragraph 2.3: Field campaigns. This would be more consistent with the title of chapter 2: Materials Methods and sites description.
Line 203: Please replace "Preview" with "Previous"
4. Estimation of volcanic CO2 from remote sensing and in situ validation
In my humble opinion this paragraph can be deleted because it adds nothing to the contents of the paper. I would recommend adding this topic only as a perspective in the conclusions.
Author Response

(The authors gave the same response as above.)

Reviewer 3 Report
Dear Authors,
I found the subject of your paper really interesting.
Overall, your paper needs an improvement of the "Discussion and Conclusion", with a more accurate comment of your results.
Please, see attached PDF for some, detailed, suggestions regarding both text and figures.
Regards,
The Reviewer

Author Response

(The authors gave the same response as above.)

Round 2
Reviewer 1 Report
Dear Authors
thanks to your replay and changes. I think that the paper has improved its potential. You have very well shown the application of your monitoring techniques.
Best wishes,